# Particles under stress: Ultrasonication causes size and recovery rate artifacts with soil derived POM, but not with microplastics.

Frederick Büks[1], Gilles Kayser[2], Antonia Zieger[1], Friederike Lang[2], Martin Kaupenjohann[1]

[1]Chair of Soil Science, Dept. of Ecology, Technische Universität Berlin, 10587 Berlin, Germany
[2]Chair of Soil Ecology, University of Freiburg, 79085 Freiburg i.Br., Germany

*Correspondence to:* Frederick Büks (frederick.bueks@tu-berlin.de)

**Abstract.** The breakdown of soil aggregates and the extraction of particulate organic matter (POM) by ultrasonication and density fractionation is a method widely used in soil organic matter (SOM) analyses. It has recently also been used for the extraction of microplastic from soil samples. However, the investigation of POM physiochemical properties and ecological functions might be biased, if particles are comminuted during the treatment. In this work, different types of POM, which are representative for different terrestrial ecosystems and anthropogenic influences, were tested for their structural stability in face of ultrasonication in a range of 0 to 500 J ml$^{-1}$. The occluded particulate organic matter (oPOM) of an agricultural and forest soil as well as pyrochar showed a significant reduction of particle size at ≥50 J ml$^{-1}$ by an average factor of 1.37±0.16 and a concurrent reduction of recovery rates by an average of 21.7±10.7 % when being extracted. Our results imply that increasing ultrasonication causes increasing retention of POM within the sedimenting phase leading to a misinterpretation of certain POM fractions as more strongly bound oPOM or part of the mineral-associated organic matter (MOM). This could e.g. lead to a false estimation of physical stabilization. In contrast, neither fresh nor weathered polyethylene (PE), polyethylene terephthalate (PET) and polybutylene adipate terephthalate (PBAT) microplastics showed a reduction of particle size or the recovery rate after application of ultrasound. We conclude that ultrasonication applied to soils has no impact on microplastic size distribution and thus provides a valuable tool for the assessment of microplastics in soils and soil aggregates.

## 1 Introduction

The mechanical disintegration of soil aggregates by use of ultrasonication following the method of Edwards and Bremner (1967a) and subsequent density fractionation of particulate organic matter is widely used in the assessment of soil organic matter (SOM) stability. This includes characteristics such as aggregate composition and stability (Edwards and Bremner, 1967b), the constitution of SOM pools (Golchin et al., 1994), the stabilization of SOM in forest ecosystems (Graf-Rosenfellner et al., 2016) and the occlusive strength of particulate organic matter (POM) (Büks and Kaupenjohann, 2016). Ultrasonication is also applied to assess quantities and qualities of anthropogenic contaminants such as microplastics (Zhang and Liu, 2018; Zhang et al., 2018).

In studies on soil carbon pools, ultrasound is applied to a soil slurry to break down soil aggregates. The disaggregation allows density fractionation of the free and occluded light fractions (fLF and oLF), which largely consist of material with densities below the fractionation medium, from the heavy fraction (HF), that has higher densities. These operational fractions largely correspond to the free particulate organic matter (fPOM), the occluded particulate organic matter (oPOM) and the mineral-associated organic matter (MOM). This organic matters are assigned to the labile, intermediate and stable carbon pool, respectively, and have turnover times of <1 year (labile) to several thousands of years (stable) (Lützow et al., 2007).

Furthermore, the extracted POM fractions may not only contain the natural but also anthropogenic components such as microplastic. Recent studies reported soil microplastic concentrations between 1 mg kg$^{-1}$ dry soil at less contaminated sites and 2 to 4 orders of magnitude above in samples from highly contaminated industrial areas (Fuller and Gautam, 2016; Rezaei et al., 2019). The agricultural application of sewage sludge, wastewater, compost as well as plastic mulching and the input of road and tire wear are discussed as important entry pathways to soils (Bläsing and Amelung, 2018). These origins of MP are characterized by a different composition of the size and shape of the extracted items (e.g. Zhang and Liu, 2018; Ding et al., 2020). In laboratory experiments, MP in the observed size range was shown to influence soil biogeochemical properties such as water holding capacity, soil structure, microbial activity and the health of soil biota, with strong dependence on the size and shape of the applied particles (de Souza Machado et al., 2018; Büks et al., 2020). Furthermore, the mobility within the soil pore space and preferential flow channels, which is crucial for the accessibility of soil microplastic to ground and surface waters, is also highly dependent on particle size (O'Connor et al., 2019; Zubris and Richards, 2005). It is therefore a very topical task for both the impact assessment of given contaminations in landscapes and the design of robust experimental setups to have extraction methods with high yield and a low alteration of microplastic size and shape.

The common method of ultrasonication is carried out with a pieco-electric converter, that uses electric energy to generate axial vibration of a sonotrode, which is dipped into a flask containing a fluid and a submerged soil sample. The oscillating sonotrode emits acoustic pulses within the fluid. In front of the shock-waves the medium is compressed, and the increased pressure causes an increased gas solubility. Behind the wave the medium relaxes and the pressure drops below the normal level leading to an explosive outgassing (Ince et al., 2001). This so called cavitation effect produces lots of exploding micro-bubbles between particles and within cavities of the soil matrix generating very local pressure peaks of 200 to 500 atm accompanied by temperatures of 4200 to 5000 K (Ince et al., 2001). It provokes the detachment of physiochemical bondings between soil primary particles and soil aggregates and, thus, causes disaggregation. Depending on the type and settings of the device, the vibration frequency can vary up to 10000 kHz, but low frequencies around 20 to 100 kHz are recommended for soil aggregate dispersion to avoid chemical alteration of OM, and the use of 40 kHz is very common (Kaiser and Berhe, 2014; Graf-Rosenfellner et al., 2018).

As an artifact of the method, ultrasonication is known to provide mechanical and thermal stress strong enough to comminute mineral particles at energy levels >700 J ml$^{-1}$ (Kaiser and Berhe, 2014). Also, the destructive influence on POM was tested in different studies and appears even at energy levels much lower than 700 J ml$^{-1}$. Without application of a solid mineral matrix, Balesdent et al. (1991) found >60 % of the POM in suspension comminuted after application of 300 J ml$^{-1}$. Amelung and Zech (1999) treated natural soils with 0 to 1500 J ml$^{-1}$ and performed a separation into size fractions of <20 µm, 20 to 250 µm and >250 µm. At ≥100 J ml$^{-1}$ POM was transferred from the >250 µm to the <20 µm fraction. In a similar manner, Yang et al. (2009) measured the mass and SOC content of sand, silt and clay sized particle fractions in natural soils using an unconventional pulse/non-pulse ultrasonication technique. The authors derived the comminution of POM at >600 J ml$^{-1}$. Oorts et al. (2005) added [13]C-enriched straw to natural soils and could show that larger amounts of POM were redistributed at 450 J ml$^{-1}$ when its degree of decomposition was higher. In conclusion, those studies consistently found a comminution of POM by ultrasonic treatment, which appears, however, at very different energy levels and is likely affected by the aggregation regime (suspended without mineral matrix, added as fPOM, occluded within natural soils), direct or indirect quantification of POM and the type of POM.

The aim of this work was to test how susceptible different POMs are to comminution by ultrasonic treatment under standardized conditions. We embedded three POMs (farm oPOM, forest oPOM and pyrochar, applied as an analog for soil black carbon and biochar amendments) and also six differently weathered microplastics (fresh and weathered low-density polyethylene (LD-PE), polyethylene terephthalate (PET) as well as polybutylene adipate terephthalate (PBAT), a common biodegradable material) into a fine sand matrix.

Then, we treated these mixtures with 0, 10, 50, 100 and 500 J $ml^{-1}$, re-extracted the organic
particles with density fractionation and measured their recovery rates and particle size
distributions. The sand matrix was used only to simulate the influence of pore space on
cavitation and, thus, our simplified approach excluded broadly varying POM–mineral
interactions resulting from aggregation processes in natural soil samples.

In advance to the treatment, the nine materials showed different mechanical stabilities. Unlike
all six types of plastic particles, the occluded POMs and the pyrochar were easily to grind
between two fingers and therefore assumed to be prone to ultrasonication. An examination of
the recent literature on microplastic extraction from soils showed that the stability of
microplastic in face of ultrasound has not been studied yet, neither with weathered nor
juvenile material. Experiments with polymer-based adsorber resins indicated fractures on
microbead surfaces after treatment with 100 J $s^{-1}$ at 40 kHz for 70 minutes (Breitbach et al.,
2002). When exposed to the environment, plastic undergoes weathering by UV radiation,
mechanical comminution, microbial decay and chemical alteration (Kale et al., 2015; Andrady
et al., 2017), which leads to embrittlement. We therefore hypothesized, that unweathered
microplastic particles will be prone to ultrasonic treatment in a degree less than weathered
microplastic and much less than pyrochar or natural oPOMs.

## 2 Material and methods

## 2.1 Preparation of POM

The farm and forest oPOMs were extracted from air-dried soil aggregates of 630 to 2000 µm in diameter sampled in 10 to 20 cm depth from an organic horticulture near Oranienburg/Brandenburg (N 52° 46' 54, E 13° 11' 50, texture Ss, $C_{org}$=49.3 g kg$^{-1}$, pH 5.8) and a spruce/beech mixed forest near Bad Waldsee/Banden-Württemberg (N 47° 50' 59, E 9° 41' 30, texture Sl4, $C_{org}$=73.2 g kg$^{-1}$, pH 3.4). The extraction was performed by use of a density fractionation in 1.6 g cm$^{-3}$ dense sodium polytungstate (SPT) solution: In 12-fold replication, 120 ml of SPT solution were added to 30 g of aggregates in a 200 ml PE bottle. The sample was stored for 1 h to allow the SPT solution to infiltrate the aggregates and was then centrifuged at 3500 G for 26 min. The floating free particulate organic matter (fPOM) was removed by use of a water jet pump and discarded. The remaining sample was refilled to 120 ml with SPT solution and sonicated for 30 sec (≈10 J ml$^{-1}$) by use of a sonotrode (Branson© Sonifier 250) in order to flaw the structure of macroaggregate (>250 µm). Then, centrifugation and removal of the oPOM were executed as for the fPOM. The gained oPOM was filtered off with an 0.45 µm cellulose acetate membrane filter, washed 3 to 5 times with 200 ml deionized water within the filter device until the rinse had an electrical conductivity of <50 µS cm$^{-1}$, removed from the filter by rinsing with deionized water, collected and gently dried for 48 h at 40°C. At the end, the oPOMs were sieved to 2000 µm, long-shaped residues were cut by a sharp knife, sieved again and pooled to one oPOM sample. The pyrogenic char sample (made from pine wood, pyrolysed at 850°C for 0.5 h by PYREG® GmbH) was dried for 24 h at 105°C, ground in a mortar and sieved to <630 µm. The microplastics (LD-PE, PET and PBAT) were made from plastic films by repeated milling (Fritsch Pulverisette 14) with liquid nitrogen and sieved to <500 µm. Then, half of each sample was weathered for 96 h at 38°C, 1000 W m$^{-2}$ (solar spectrum, 280 to 3000 nm) and a relative air humidity of 50 % following DIN EN ISO 4892-2/3, which is the international industry standard for testing artificial weathering of polymere-based materials (Pickett, 2018).

## 2.2 Mechanical stress treatment

In order to test their stability against ultrasonication, the nine POM types (farm and forest oPOM and pyrochar as well as fresh and weathered LD-PE, PET and PBAT) were each exposed in triplicates to different mechanical stress levels (0, 10, 50, 100 and 500 J ml$^{-1}$). The treatment with 0 J ml$^{-1}$ was used as a control with no mechanical agitation and 10 J ml$^{-1}$ represents a gentle stimulation, which is suggested not to disaggregate soil structure (Kaiser and Berhe, 2014). Macroaggregates are prone to 50 J ml$^{-1}$, and 100 to 500 J ml$^{-1}$ mark the range of microaggregate disaggregation, as many studies stated full disaggregation of soils after application of ~500 J ml$^{-1}$ (Kaiser and Berhe, 2014). Larger values were ruled out,

although some studies applied energy levels above 500 J ml$^{-1}$, like Pronk et al. (2011) who could show that silt-sized microaggregates were not dispersed at energy levels ≤800 J ml$^{-1}$. However, small microaggregates often contain little or no POM (Tisdall, 1996), and energies >710 J ml$^{-1}$ cause physical damage on mineral particles (Kaiser and Berhe, 2014). Therefore we focus on the range of 0 to 500 J ml$^{-1}$ as a safe space for the extraction of POM with no other known artifacts.

We chose acid-washed and calcinated fine sand to simulate the soil mineral matrix. This texture can be easily suspended by ultrasonication (coarse sand cannot), has a low tendency to coat POM or coagulate (like clay does) and shows a fast sedimentation when the sample is centrifuged. Fine sand, moreover, represents soils that originated from Weichselian sanders or aeolian sand deposition. In this methodical paper, our aim, however, was not to simulate a set of soil textures, but to have a proof of concept to find out if natural or artificial POM is damaged by ultrasonication. Then, quantities of 1 % w/w POM, and 0.5 % w/w in case of the oPOMs, were embedded into the fine sand matrix.

These artificial soils (each 20 g) were stored in 100 ml of 1.6 g cm$^{-3}$ dense SPT solution for 1 h in 200 ml PE bottles, that did not show measurable release of plastic fragments due to sonication in preliminary tests with a pure fine sand matrix (data not shown). Mechanical stress was applied by use of a sonotrode (Branson© Sonifier 250) as described by Büks and Kaupenjohann (2016). The sonication times corresponding to 0, 10, 50, 100 and 500 J ml$^{-1}$ were determined by means of the sonotrode's energy output calculated following North (1976). After the ultrasonic treatment, samples were centrifuged at 3500 G for 26 min. The floated POM was removed by use of a water-jet pump, separated and cleaned by rinsing with deionized water on a 0.45 µm cellulose acetate membrane filter until the electrical conductivity of the rinse went below 50 µS cm$^{-1}$, and then lyophilized.

## 2.3 Determination of recovery rates

After lyophilization, the recovery rate $R=m_t m_0^{-1}$ was determined by weighing and described as ratio of the recovered POM mass after treatment ($m_t$) to the initial POM mass ($m_0$) for all POM types and energy levels. The recovery rate of a certain energy level is assumed significantly different to the 0 J ml$^{-1}$ level, if a pairwise t-test results in a $p<0.05$ (Table 1).

## 2.4 Measurement of particle sizes

All samples continued to be used for particle sizing. After pre-trials have shown that mainly the hydrophobic particles (microplastics and pyrochar) coagulated in distilled water, aggregation was avoided by suspension in 0.1 % w/v Tween© 20 detergent solution and vortexing following Katija et al. (2017). 30 to 100 mg of POM were suspended in 500 ml 0.1 %

Tween© 20 solution and size classified with a QICPIC image analysis device (Sympatec
GmbH, Clausthal-Zellerfeld, Germany) using a modified method from Kayser et al. (2019).
Counts were grouped into 34 size classes from <5.64 µm to 1200–1826.94 µm and plotted as
cumulative histograms of each replicate and their mean values (Fig. 1a and 1b). As the
primary criterion for the reduction in particle size, the first 10 % and 50 % quantile (median)
values were compared by pairwise t-test between 0 J ml$^{-1}$ and each other energy level,
respectively. As particle size reduction could be significant but still marginal in case of a low
variance between parallels and a low grade of comminution at the same time, the averaged
comminution factor (CF) was introduced. It is defined as

$$CF = \frac{\sum_i \left(\frac{x_{0,i}}{x_i}\right)}{i} \quad (1)$$

with i the number of parallels, $x_{0,i}$ the quantile value of the 0 J ml$^{-1}$ energy level and $x_i$ the
value of the compared energy level. A sample is then assumed significantly different to the
0 J ml$^{-1}$ control and not marginal, if the p-value given by the t-test is <0.05 and the
comminution factor is >1.1 for the 10 % quantile, the median or both, while its standard
deviation is sd<|CF-1|. (Table 2)

## 2.5 Organic matter balance

A second set of triplicates of pyrochar and farm soil oPOM were treated similarly at 0 and
500 J ml$^{-1}$ to balance the complement of the recovered POM. For this purpose, the C
concentration within the lyophilized sediment was measured by use of a CNS analyzer and
converted to POM mass by use of the C content (%) of the respective organic matter. In
addition, the mass gain of the cellulose acetate filters was measured after rinsing the sample
and drying the filter at 70°C for 24 hours. The DOC concentration of the filtrate was measured
and converted to DOM by use of an assumed 50 % C content. The difference of these and
the recovered fractions compared to the initial weight of organic particles is termed the
balance loss during the extraction procedure. (Table 3)

## 3 Results

### 3.1 Resulting recovery rates

All microplastic samples (LD-PE, PET and PBAT) show a constantly high recovery rate of about 97.1±2.5 % in average over the whole range of applied energy levels. In sharp contrast, all other samples were decreasingly recovered along with increasing energy levels. Farmland POM, forest POM and pyrochar showed significant differences to the 0 J ml$^{-1}$ treatment at ≥10 J ml$^{-1}$, ≥100 J ml$^{-1}$ and ≥ 100 J ml$^{-1}$, respectively. (Table 1)

**Table 1:** Recovery rates of natural POMs and microplastics from after ultrasonic treatment with 0, 10, 50, 100 and 500 J ml$^{-1}$ (n=3). The (w) marks weathered plastics, mv mean value and sd standard deviation. Bold numbers are significantly different from the 0 J ml$^{-1}$ treatment by p<0.05.

| sample | 0 J ml$^{-1}$ mv sd | 10 J ml$^{-1}$ mv sd | recovery rate [%w/w] 50 J ml$^{-1}$ mv sd | 100 J ml$^{-1}$ mv sd | 500 J ml$^{-1}$ mv sd |
|---|---|---|---|---|---|
| farm oPOM | 95.0 ± 2.3 | **80.8 ± 4.5** | **73.2 ± 6.1** | **72.3 ± 2.8** | **51.6 ± 7.2** |
| forest oPOM | 89.3 ± 5.4 | 79.0 ± 5.1 | 76.9 ± 8.4 | **67.8 ± 3.6** | **48.7 ± 5.4** |
| pyrochar | 93.5 ± 10.1 | 84.6 ± 6.1 | 78.1 ± 2.5 | **74.3 ± 1.9** | **63.8 ± 3.1** |
| LD-PE | 96.9 ± 1.2 | 97.3 ± 1.0 | 95.8 ± 6.7 | 99.9 ± 1.9 | 99.2 ± 1.6 |
| LD-PE (w) | 93.9 ± 3.4 | 96.5 ± 1.2 | 96.6 ± 1.5 | 98.9 ± 3.0 | 97.8 ± 1.7 |
| PET | 98.6 ± 2.5 | 94.0 ± 1.6 | 98.7 ± 2.5 | 98.5 ± 2.0 | 94.3 ± 1.3 |
| PET (w) | 96.2 ± 2.5 | 95.4 ± 3.0 | 97.0 ± 2.0 | 95.5 ± 1.0 | 96.4 ± 3.3 |
| PBAT | 99.6 ± 2.5 | 99.5 ± 0.9 | 90.9 ± 13.8 | 98.3 ± 3.6 | 98.2 ± 0.9 |
| PBAT (w) | 97.7 ± 0.9 | 99.3 ± 1.9 | 96.8 ± 1.6 | 96.6 ± 1.7 | 99.3 ± 1.9 |

### 3.2 POM size distribution

None of the plastics shows a significant reduction of particle size due to ultrasonic treatment within the 10 % and 50 % quantile. In contrast, at ≥100 J ml$^{-1}$ the particle size of farm and forest oPOM was significantly reduced compared to the 0 J ml$^{-1}$ treatment in both quantiles. Ultrasonic treatment also causes a significant comminution of pyrochar, but of mainly the smaller fraction indicated by the 10 % quantile, which appeared at ≥50 J ml$^{-1}$ and is only interrupted due to an outlier at 100 J ml$^{-1}$. The 50 % quantile data (median) remain insignificant. (Fig. 1a and 1b , Table 2)

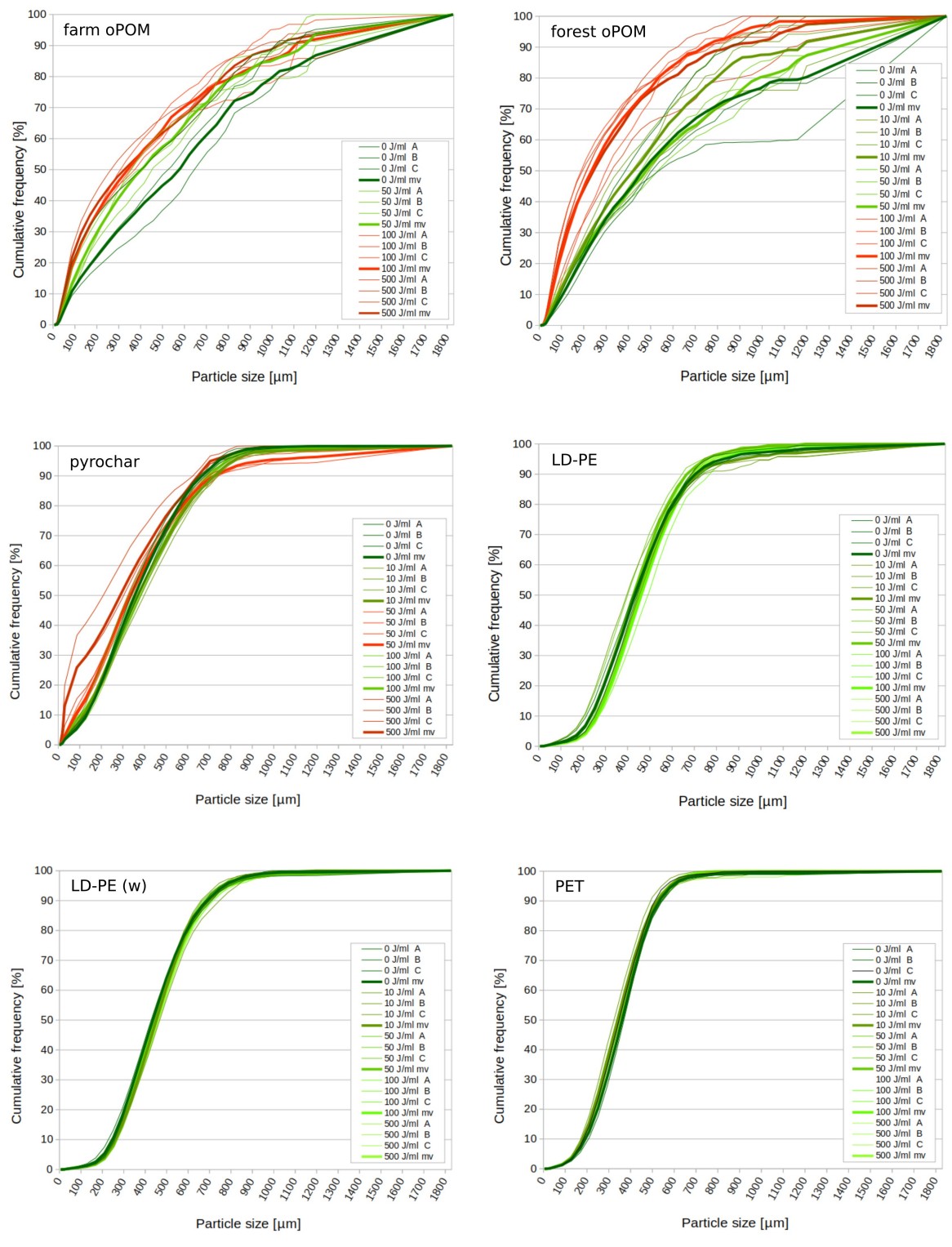

**Figure 1a:** Particle size distribution of natural POMs and microplastics after ultrasonic treatment with 0, 10, 50, 100 and 500 J ml$^{-1}$ (n=3: A, B, C). The (w) marks weathered plastics. Green graphs are similar to the 0 J ml$^{-1}$ treatment, red graphs significantly different by p<0.05 and comminution factor >1.1. Bold lines represent mean values (mv).

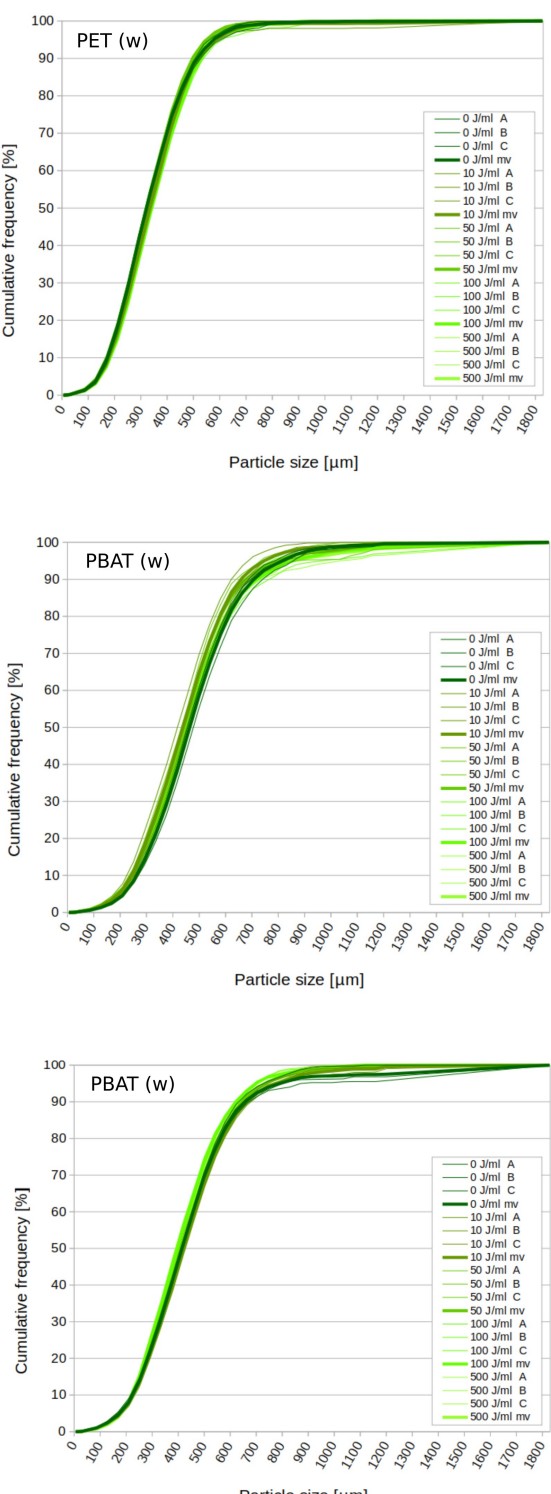

**Figure 1b:** Particle size distribution of microplastics after ultrasonic treatment with 0, 10, 50, 100 and 500 J ml$^{-1}$ (n=3: A, B, C). The (w) marks weathered plastics. Green graphs are similar to the 0 J ml$^{-1}$ treatment (p≥0.05 or comminution factor ≤1.1). Bold lines represent mean values (mv).

**Table 2:** Particle size distribution (10 % and 50 % quantile) and comminution factor of natural POMs and microplastics after ultrasonic treatment with 0, 10, 50, 100 and 500 J ml$^{-1}$ (n=3). The (w) marks weathered plastics, mv mean value and sd standard deviation. Bold numbers are significantly different from the 0 J ml$^{-1}$ treatment by p<0.05 and comminution factor >1.1.

| POM type | J/ml | size distribution | | | | comminution factor | | | |
| | | 10% quantile | | 50% quantile | | 10% quantile | | 50% quantile | |
| | | mv | sd | mv | sd | mv | sd | mv | sd |
| --- | --- | --- | --- | --- | --- | --- | --- | --- | --- |
| | 0 | 82.90 ± 9.46 | | 561.33 ± 72.98 | | 1.00 ± 0.00 | | 1.00 ± 0.00 | |
| | 10 | N/A | | N/A | | N/A | | N/A | |
| farm oLF | 50 | 72.31 ± 15.39 | | **401.40 ± 47.86** | | **1.17 ± 0.15** | | 1.17 ± 0.34 | |
| | 100 | **53.40 ± 2.61** | | **344.64 ± 33.40** | | **1.56 ± 0.26** | | **1.56 ± 0.23** | |
| | 500 | **47.21 ± 2.46** | | **365.57 ± 52.18** | | **1.76 ± 0.21** | | **1.76 ± 0.23** | |
| | 0 | 108.08 ± 17.40 | | 476.26 ± 79.01 | | 1.00 ± 0.00 | | 1.00 ± 0.00 | |
| | 10 | 91.71 ± 11.04 | | 422.27 ± 68.13 | | 1.19 ± 0.27 | | 1.17 ± 0.36 | |
| forest oLF | 50 | 84.92 ± 16.97 | | 485.08 ± 41.44 | | **1.28 ± 0.09** | | 0.98 ± 0.14 | |
| | 100 | **60.48 ± 16.40** | | **233.11 ± 58.78** | | **1.87 ± 0.55** | | **2.18 ± 0.80** | |
| | 500 | **55.49 ± 13.01** | | **244.41 ± 70.33** | | **1.98 ± 0.28** | | **2.02 ± 0.48** | |
| | 0 | 130.33 ± 6.33 | | 355.79 ± 16.19 | | 1.00 ± 0.00 | | 1.00 ± 0.00 | |
| | 10 | 119.09 ± 16.07 | | 369.18 ± 39.01 | | 1.10 ± 0.11 | | 0.97 ± 0.15 | |
| pyrochar | 50 | **81.39 ± 10.07** | | 333.41 ± 9.59 | | **1.62 ± 0.25** | | 1.07 ± 0.08 | |
| | 100 | 103.37 ± 33.73 | | 371.92 ± 19.99 | | 1.34 ± 0.38 | | 0.96 ± 0.09 | |
| | 500 | **31.18 ± 11.70** | | 284.35 ± 67.85 | | **4.59 ± 1.67** | | **1.30 ± 0.28** | |
| | 0 | 235.15 ± 19.46 | | 433.21 ± 9.18 | | 1.00 ± 0.00 | | 1.00 ± 0.00 | |
| | 10 | 236.54 ± 29.80 | | 432.25 ± 31.43 | | 1.00 ± 0.06 | | 1.01 ± 0.06 | |
| LD-PE | 50 | 237.80 ± 28.51 | | 425.20 ± 26.47 | | 1.01 ± 0.20 | | 1.02 ± 0.08 | |
| | 100 | 263.23 ± 6.87 | | 463.10 ± 24.59 | | 0.89 ± 0.05 | | 0.94 ± 0.03 | |
| | 500 | 266.29 ± 5.32 | | 454.22 ± 9.98 | | 0.88 ± 0.06 | | 0.95 ± 0.01 | |
| | 0 | 245.69 ± 15.39 | | 435.02 ± 6.41 | | 1.00 ± 0.00 | | 1.00 ± 0.00 | |
| | 10 | 260.20 ± 5.64 | | 451.72 ± 16.36 | | 0.94 ± 0.04 | | 0.96 ± 0.03 | |
| LD-PE (w) | 50 | 265.51 ± 1.55 | | **451.20 ± 6.71** | | 0.93 ± 0.06 | | 0.96 ± 0.03 | |
| | 100 | 253.61 ± 7.67 | | 442.70 ± 3.57 | | 0.97 ± 0.08 | | 0.98 ± 0.02 | |
| | 500 | 262.94 ± 3.25 | | **458.59 ± 4.03** | | 0.93 ± 0.06 | | 0.95 ± 0.02 | |
| | 0 | 193.66 ± 11.91 | | 360.74 ± 11.96 | | 1.00 ± 0.00 | | 1.00 ± 0.00 | |
| | 10 | 180.15 ± 7.97 | | 339.89 ± 13.84 | | 1.08 ± 0.12 | | 1.06 ± 0.07 | |
| PET | 50 | 179.69 ± 5.09 | | 344.78 ± 7.76 | | 1.08 ± 0.09 | | 1.05 ± 0.06 | |
| | 100 | 162.59 ± 29.24 | | **341.00 ± 1.94** | | **1.21 ± 0.19** | | 1.06 ± 0.04 | |
| | 500 | 181.14 ± 7.12 | | 344.70 ± 6.93 | | 1.07 ± 0.08 | | 1.05 ± 0.04 | |
| | 0 | 171.89 ± 5.20 | | 321.46 ± 4.19 | | 1.00 ± 0.00 | | 1.00 ± 0.00 | |
| | 10 | 186.44 ± 11.60 | | 332.81 ± 7.80 | | 0.92 ± 0.07 | | 0.97 ± 0.01 | |
| PET (w) | 50 | 172.80 ± 7.98 | | 324.73 ± 7.55 | | 1.00 ± 0.08 | | 0.99 ± 0.04 | |
| | 100 | **182.74 ± 0.80** | | **340.28 ± 7.11** | | 0.94 ± 0.03 | | 0.95 ± 0.03 | |
| | 500 | 157.67 ± 25.54 | | 331.51 ± 9.52 | | 1.11 ± 0.18 | | 0.97 ± 0.04 | |
| | 0 | 263.19 ± 6.13 | | 464.20 ± 11.93 | | 1.00 ± 0.00 | | 1.00 ± 0.00 | |
| | 10 | 243.05 ± 15.60 | | 437.71 ± 18.57 | | 1.09 ± 0.08 | | 1.06 ± 0.04 | |
| PBAT | 50 | **240.26 ± 6.80** | | 441.55 ± 9.41 | | 1.10 ± 0.04 | | 1.05 ± 0.05 | |
| | 100 | 246.75 ± 5.27 | | **455.51 ± 5.37** | | 1.07 ± 0.02 | | 1.02 ± 0.04 | |
| | 500 | **242.52 ± 3.78** | | 452.18 ± 11.85 | | 1.09 ± 0.04 | | 1.03 ± 0.05 | |
| | 0 | 223.53 ± 6.06 | | 413.87 ± 4.60 | | 1.00 ± 0.00 | | 1.00 ± 0.00 | |
| | 10 | 225.56 ± 6.97 | | **423.06 ± 2.81** | | 0.99 ± 0.06 | | 0.98 ± 0.02 | |
| PBAT (w) | 50 | 225.22 ± 2.92 | | 414.68 ± 8.41 | | 0.99 ± 0.04 | | 1.00 ± 0.02 | |
| | 100 | 220.13 ± 1.97 | | **396.85 ± 6.20** | | 1.02 ± 0.03 | | 1.04 ± 0.03 | |
| | 500 | 224.71 ± 5.53 | | 404.80 ± 12.40 | | 1.00 ± 0.03 | | 1.02 ± 0.04 | |

## 3.3 Mass loss

The treatment of pyrochar triplicates with 500 J ml$^{-1}$ resulted in a recovery rate of 54.3±5.2 % after density fractionation. In turn, 34.9±3.7 % of the POM remained in the sediment, 0.6±0.1 % into the DOM fraction and <0.5 % onto the filter, leading to a balance loss of 10.2±2.1 % (Table 3). The respective data of farm oPOM are 54.6±1.9 %, 20.3±3.1 %, 5.1±0.2 %, <0.5 % and 20.0±1.5 %. Samples treated with 0 J ml$^{-1}$ instead showed a significantly higher recovery rate and lower retention compared to the 500 J ml$^{-1}$ samples. In contrast, the balance loss remained constant between 0 and 500 J ml$^{-1}$.

**Table 3:** Mass balance that indicates the fate of OM fractions during the ultrasonication/density fractionation treatment. Bold numbers indicate differences with $p < 0.05$ after t-test between the 0 and 500 J ml$^{-1}$ variant (n=3).

| POM (energy level) | recovery (%) | retention (%) | filter (%) | DOM (%) | mass loss (%) |
|---|---|---|---|---|---|
| pyrochar (0 J ml$^{-1}$) | **79.6±3.6** | **8.7±0.3** | <0.5 | **0.3±0.0** | 11.4±3.4 |
| pyrochar (500 J ml$^{-1}$) | **54.3±5.2** | **34.9±3.7** | <0.5 | **0.6±0.1** | 10.2±2.1 |
| farm oPOM (0 J ml$^{-1}$) | 64.8±6.9 | **8.3±0.2** | <0.5 | **2.7±0.0** | 24.1±6.8 |
| farm oPOM (500 J ml$^{-1}$) | 54.6±1.9 | **20.3±3.1** | <0.5 | **5.1±0.2** | 20.0±1.5 |

## 4 Discussion

Our experiments indicate that soil derived oPOM and pyrochar embedded into a fine sand matrix are prone to comminution by ultrasonic treatment at energy levels of ≥50 J ml$^{-1}$. These values are well below the 300 to 750 J ml$^{-1}$ given in the literature for the complete disaggregation of various soils (Amelung and Zech, 1999; Oorts et al., 2006; Yang et al., 2009), namely in the range of values given for the destruction of macroaggregates (Amelung and Zech, 1999; Kaiser and Berhe, 2014). This underpins the former implications by some authors that ultrasonic treatment could lead to particle size artifacts. Microplastic, in contrast, shows a constant particle size distribution over all energy levels and seems to resist ultrasonication within the tested range of 0 to 500 J ml$^{-1}$. The recovery of microplastics also shows a constantly high rate of nearly 100 %, which is not affected by the applied energy. In sharp contrast, the recovery rates of soil derived POMs and pyrochar decreased with increasing energies from 95.0 to 78.6 % to 63.8 to 35.8 %, which became significant at 50 to 100 J ml$^{-1}$ and therefore is quite parallel to observed size reduction.

The concurrent decrease of particle size and recovery rate of soil derived POMs and pyrochar and its absence after ultrasonic treatment of microplastics might indicate a causal relationship of these measures. The underlying process, however, has not been studied before. We assume a mechanism that prevents POM from density fractionation. This effect appeared in our experiment from energies around 50 J ml$^{-1}$ with the beginning destruction of oPOM. As mentioned in Ince et al. (2001) and confirmed in Kaiser and Berhe (2014), ultrasonication induced high temperature may reduce total C content due to oxidative reactions, but the balance loss, constant between 0 and 500 J ml$^{-1}$ in both pyrochar and farm oPOM, implies that there is no burning of organic matter due to ultrasound treatment. Also the formation of large amounts of water-soluble molecules and colloids could be ruled out in our experiment. The recovery rate decreases in the same degree as the retention in the sediment increases when ultrasound is applied, while filter residues and lost DOM, which doubled on a low level, play a minor role. Extreme thermal conditions occuring during ultrasoincation, however, may explain the increased retention of POM within the sediment. Sparse data on molecular alteration of organic materials due to ultrasonication showed the transformation of lignin, a major constituent of plant cell walls. One hour of treatment caused the formation of a high molecular weight fraction of about 35% of the lignin content with molecular weights increased by the 450-fold (Wells et al., 2013). This may also increase the density of lignin and ligninoid fractions in soil POM towards the density of the fractionation medium and reduce their recovery rate.

As a consequence of the reduction of the recovery rate, farmland, forest and pyrochar POMs remain within a sandy matrix the stronger they are treated by ultrasound. If these findings are applied to ultrasonication/density fractionation of natural soils, not only an increasing number of particle size artifacts can be expected, but also the extraction of occluded POM is

increasingly hindered at a certain energy level. After each extraction step, parts of the released oPOM remain within the sedimenting fraction, a carry-over artifact. This leads to an underestimation of the extracted oPOM fractions and an overestimation of the mineral-associated organic matter fraction (MOM), that natural part of the soil organic matter (SOM), which is adsorbed on mineral surfaces of the heavy fraction and mainly assumed to be molecular. According to our data, a reduction of recovery rates would appear at 10 J ml$^{-1}$ in farmland soils and 100 J ml$^{-1}$ in forest soils as well as at 100 J ml$^{-1}$ when extracting pyrochar particles. Thus, the artifact would affect the extraction of oPOM from microaggregates of all samples and also the extraction of oPOM from macroaggregates in farmland soils. However, further research has to elucidate, if these results can be applied to natural soil samples.

An overestimation would have an impact e.g. on the assessment of operationally defined carbon pools within landscapes: POM is assigned to carbon pools with turnover times orders of magnitude shorter then MOM, that endures hundreds of years. Malquantifications of these pools, such as counting POM to the MOM as implied by this work, would have influence on e.g. the estimation of SOM decomposition and $CO_2$ emissions from land-use change. Carrying-over SOM from little to highly decomposed fractions also could alienate genuine C:N ratios, which strongly differ between the functional carbon pools (Wagai et al., 2009). In respect to coming experiments, comminution and reduced recovery rate of the oPOM can possibly be avoided by not exceeding the energy levels mentioned here – or by determining a specific energy cut-off for each natural soil in preliminary studies. Regarding the application of higher energy levels, detailed investigation on the underlying mechanism are necessary to give such recommendations.

Microplastic particles, whether they are weathered following DIN ENISO4892-2/3 or pristine, are not prone to disruption by ultranonic treatment and its recovery rates are stable in a wide range of energy levels. We therefore assume that there will be significantly less carry-over of particles due to comminution when extracting microplastics from soils with ultrasonication/density fractionation. In consequence, the extractive performance is higher and subsequent particle size measurements give more valid information about the original particle size spectrum compared to the measurement of farmland, forest and pyrochar POM. This is a positive sign for research on soil microplastic, however, it does not mean that microplastic will be fully extracted from soils by this method. Soil microplastics appear within a wide range of sizes between some nanometers and its upper limit of 5 mm by definition. Their smallest part, fibers and microfragments produced by physical, chemical and biological erosion within the soil, might also be affected by chemical alteration due to both weathering and ultrasonication causing enhanced retention in the sedimenting fraction. Although we have introduced billions of tons of microplastics into ecosystems since the 1950s (Thompson et al., 2009; Geyer et al., 2017), there are still problems in producing microplastic fragments

<100 µm on a laboratory scale with adequate use of time and material to perform experiments
within this size range.

## 5 Conclusion

Unlike weathered and fresh PE, PET and PBAT microplastic, soil derived POMs like occluded POM from farm and forest soils and pyrochar concurrently show comminution and a reduced recovery rate after ultrasonication and subsequent extraction from a sandy matrix. Applied to natural soils, parts of the farmland, forest and pyrochar POM remain within the sedimenting fraction and can be misinterpreted as more strongly bound oPOM or MOM. An overestimation as shown in this study might lead to fundamentally different interpretations of physical protection of SOM, functional carbon pools and the expected mineralization rates in consequence of e.g. land-use change. On the contrary, the extraction of microplastics neither causes additional retention of particles nor alienates the particle size spectrum due to ultrasonic-driven comminution. We conclude that density fractionation in combination with ultrasonication is an appropriate tool for analyzing occlusion of microplastics within soil aggregates and studying the size distribution of particulate microplastics.

## Author contribution

Frederick Büks developed the experimental concept, extracted all samples and prepared the manuscript. Gilles Kayser performed the particle size analysis. Antonia Zieger supported the development of the experimental concept. Martin Kaupenjohann and Friederike Lang supervised the whole study.

## Acknowledgement

Many thanks to Zoltán Mátra, who kindly helped us to conduct the QICPIC analysis.

## Competing interests

The authors declare that they have no conflict of interest.

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
