# Peer review of "Particles under stress: Ultrasonication causes size"

_Biogeosciences, 2020_

## Referee Comment (RC1) · Anonymous Referee #1 · 30 Jun 2020

General Comments The paper analyses the effects of ultrasonication on the particle size distribution of soil organic particles and microplastics. Both is highly relevant for communities dealing with soil organic matter fractionation and with separation and analyses of microplastics in soil, which is an emerging "hot topic" in soil science. However, the methodological approach needs much more justification, the chosen parameters are just stated and appear randomly selected. If this paper wants to set the tone and serve as a cornerstone in microplastic separation from soil, which is a crucial step for their further analytics and quantification, the reader deserves detailed explanation and justification about, for example, the chosen energy levels, the composition of the artificial soil, or the different particle sizes of organic particles, pyrochar, and plastics.

I detailed several issues in my specific comments that need to be addressed. The discussion lacks in large parts to put the experimental settings and findings into soil ecological context. At several places, the authors formulate very general statements within the discussion, which are, from my perspective, less sufficiently backed up by the data. Please see my specific comments. Furthermore, in the light of only 1.5 pages of discussion, which is about one page in print, the manuscript seems to be less suitable to be published as a full research paper. Instead, I recommend resubmitting a revised version of the manuscript as a technical note or short communication.

Specific Comments Introduction Line 11: delete "some" Line 36: ultrasound is applied to a soil slurry by using a sonotrode Lines 36-37: "light" and "heavy" needs to be explained here Lines 38-42: split into two sentences Line 45: define "extractive performance" and give more reasoning why research in the field of soil contamination with microplastic is crucial Line 49: "Büks et al., in review" is not a valid reference Line 50: sentence is missing that connects this paragraph with the paragraph before Line 63: punctuation mark is not at the correct spot Lines 91-93: Why do you assume this? You need to justify your assumption; otherwise, it is not a hypothesis. The phrase "we were curious" is a weak justification for doing research, provide here a solid hypotheses driven reasoning and provide prove for your claim that this has not studied before, based on what research (literature search?) you conclude this?

Materials and Methods Lines 114-119: hy did you use different particle sizes for soil POM, char POM, and plastic POM, please justify because different particle sizes might affect the outcome Lines 119-121: the weathering approach is not clear to me, justify and explain in more detail, and according to Table 1 and 2 only microplastic samples were weathered, please clarify this here Line 125: why this stress levels, please justify your selection and why do you use J/ml and not the more common J/cm3 unit? Line 126: why 1% and 0.5%, please justify these amounts Line 127: if you want to simulate the soil matrix, why did you used only fine sand and not a more heterogeneous mixture?

Results Line 171: two times 100 J/ml

Figure 1 and Table 1 present the same data, so they are redundant, please remove Figure 1

For Table 1 and 2, from my prospective, a paired t-test requires independent samples but your samples are not independent (POM forest is from one soil, LD-PE from one plastic film, for example) based on that you can just state an increase or a decrease or you go for mean values (per energy amount) from farm POM, arable POM, and pyrochar ("natural POM", n = 3) and mean values (per energy amount) from all plastic samples ("microplastic POM", n = 6, this group could be further subdivided into weathered or not weathered), then energy amount or energy amount/ size ditribution can serve as factors in an ANOVA analysis,

Captions for Figures 2 and b: A, B, and C as well as mv need to be explained

Discussion Lines 181-195: this needs to be discussed in the light of the experimental settings, the artificial soil just contained POM and fine sand, how can these findings be applied to soils with much more clay or iron oxides?  what would be a step forward to avoid such effects?  Line 197: I do not really see a causal relationship here, please clarify Line 199: this would mean that the fine sand particles form associations with small organic particles but I do not see any evidence for this or a paper cited that describes such phenomena, a reason might be that the density of natural POM is changing because of stronger solubilization processes of smaller particles in density solutions Line 200: needs to be "specific surface area in cm2/g" Line 203: again, you only have mineral surface of fine sand particles, which are usually less involved in organic matter-mineral associations, this needs to be clarified on a mechanistic level using appropriate literature if no own data can be used Lines 206-207: why that? please provide more detailed explanations on potential effects on fPOM without any stress from sonication Lines 208-217: again, this is a very general statements but need to be seen in the perspective of your specific experimental settings, and what experiments

would be necessary to get more general information Line 222: define "phenomenal influence" Lines 218-225: again, any recommendations how such effects could be minimized during fractionation Line 226: again, very general statement, define "plastic" Line 227: what about above 500 J/ml? and I recommend to avoid statements like "no carry-over", for such a bold statement the data are not sufficient

---

## Author Comment (AC1) · 16 Jul 2020

Dear referee #1.

Many thanks for your mindful proofreading, the precise and very helpful comments. It has helped us to see some points which still need clarification. In the following, we want to explain how we propose to adjust our article based on the reviewer's comments and also explain why in some cases we do not agree with the reviewer's proposed changes. We also think that due to the added explanations the work exceeds the frame of a technical note or short communication.

Abstract

Line 11: delete "some" [1] Done.

Introduction

Line 36: ultrasound is applied to a soil slurry by using a sonotrode and Lines 36-37: "light" and "heavy" needs to be explained here [2] We adjusted the Lines 36-37 "In studies on soil carbon pools, ultrasound is applied to a soil slurry to break down soil aggregates." [3] and added the explanation of LF and HF (Line 38): "This disaggregation allows density fractionation of the free and occluded light fractions (fLF and oLF), which largely consist of material with densities below the fractionation medium, from the heavy fraction (HF), that has higher densities." [4] Furthermore, "... and subsequent density fractionation of particulate organic matter ..." is added to Line 29 to introduce the fact that density fractionation is an integral part of the method.

Lines 38-42: split into two sentences [5] Done.

Line 45: define "extractive performance" and give more reasoning why research in the field of soil contamination with microplastic is crucial. and Line50: sentence is missing that connects this paragraph with the paragraph before [6] We propose to split the paragraph at line 42 and rephrase and complement the following part: "Furthermore, the extracted POM fractions may not only contain the natural but also anthropogenic components such as microplastic. Recent studies reported soil microplastic concentrations between 1 mg kg-1 dry soil at less contaminated sites and 2 to 4 orders of magnitude larger than in samples from highly contaminated industrial areas (Fuller and Gautam, 2016; Rezaei et al., 2019). The agricultural application of sewage sludge, wastewater, compost as well as plastic mulching and the input of road and tire wear are discussed as important entry pathways to soils (Bläsing and Amelung, 2018). These origins of MP are characterized by a different composition of the size and shape of the extracted items (e.g. Zhang and Liu, 2018; Ding et al., 2020). In laboratory experiments, MP in the observed size range was shown to influence soil biogeochemical properties such as

water holding capacity, soil structure, microbial activity and the health of soil biota, with strong dependence on the size and shape of the applied particles (de Souza Machado et al., 2018; Büks et al., 2020). Furthermore, the mobility within the soil pore space and preferencial flow channels, which is crucial for the accessibility of soil microplastic to ground and surface waters, is also highly dependent on particle size (O'Connor et al., 2019; Zubris and Richards, 2005). It is therefore a very topical task for both the impact assessment of given contaminations in landscapes and the design of robust experimental setups to have extraction methods with high yield and a low alteration of microplastic size and shape."

Line 49: "Büks et al., in review" is not a valid reference [7] Now it is: Büks, F., van Schaik, N. L., and Kaupenjohann, M.: What do we know about how the terrestrial multicellular soil fauna reacts to microplastic?, SOIL, 6, 245–267, https://doi.org/10.5194/soil-6-245-2020, 2020.

Line63: punctuation mark is not at the correct spot [8] Done.

Lines 91-93: Why do you assume this? You need to justify your assumption; otherwise, it is not a hypothesis. The phrase "we were curious" is a weak justification for doing research, provide here a solid hypotheses driven reasoning and provide prove for your claim that this has not studied before, based on what research (literature search?) you conclude this? [9] We really agree with the author's point, that we did not provide a sufficient hypothesis and therefore propose to add a new paragraph after Line 91: "In advance to the treatment, the nine materials showed different mechanical stabilities. Unlike all six types of plastic particles, the occluded POMs and the pyrochar were easily to grind between two fingers and therefore assumed to be prone to ultrasonication. An examination of the recent literature on microplastic extraction from soils showed that the stability of microplastic in face of ultrasound has not been studied yet, neither with weathered nor juvenile material. Experiments with polymer-based adsorber resins indicated fractures on microbead surfaces after treatment with 100 J s-1 at 40 kHz for 70 minutes (Breitbach et al., 2002). When exposed to the environment, plastic

undergoes weathering by UV radiation, mechanical comminution, microbial decay and chemical alteration (Kale et al., 2015; Andrady et al., 2017), which leads to embrittlement (Quelle). We therefore hypothesized, that unweathered microplastic particles will be prone to ultrasonic treatment in a degree less than weathered microplastic and much less than pyrochar or natural oPOMs."

Materials and Methods

Lines 114-119: why did you use different particle sizes for soil POM, char POM, and plastic POM, please justify because different particle sizes might affect the outcome. [10] We propose to add the following explanation to the discussion. "The different sizes of the particles are caused by their origin. Data show, that a high percentage of MP in soils is <250 $\mu$m (e.g. Zhang and Liu, 2018). However, in laboratory PE, PET and PBAT are not comminutable to those sizes in larger extent with a passable expenditure of time by cryo-milling (several days of milling with permanent application of liquid N2) or any other known method. Alternatively, an extraction of MP from soils would not lead to pure or unweathered material and requires the treatment of tens of kg of soil. Pyrochar, in contrast, is comminuted to a similar size spectrum as the MP, but with slightly higher proportion of small particles, only by gentle pestling. The oPOM samples were extracted to represent the size spectrum in natural soils and have a higher proportion of both small and large particles compared to MP. However, from our point of view this would not alter the quality of the results: Based on the theory of statistical brittle fracture (which is also applied to soil aggregates by Braunack et al., 1979), particles of the same material are statistically more fragile faced to mechanical stress if they have larger size and, thus, a higher probability of flaws within their structure. We therefore assume that by use of particle size spectra similar to that of the plastic particles, pyrochar and oPOMs would show a more distinct degree of comminution. On the other hand, smaller MP is not expected to be comminuted as larger particle remain intact. The qualitative statement, that natural POMs/pyrochar are more prone to mechanical stress than MP and size/recovery artifacts are highly probable, would not be altered."

Lines 119-121: the weathering approach is not clear to me, justify and explain in more detail, and according to Table 1 and 2 only microplastic samples were weathered, please clarify this here. [11] We propose to add to Line 121: "..., which is the international industry standard for testing artificial weathering of polymere-based textiles and coatings." and "This approach is applied for pre-treatment of MP in current experiments knowing that also microbial processes might play a role in weathering of soil MP (Kale et al., 2015). However, there is no established method including this, yet." to the discussion section.

Line 125: why this stress levels, please justify your selection and why do you use J/ml and not the more common J/cm3 unit? and Line 227 (Discussion): what about above 500 J/ml? [12] Both units J ml-1 and J cm-3 are common. If it is really wished, we will change it to J cm-3. [13] For justification of the chosen energy levels, we propose to insert the following text after Line 125: "The treatment with 0 J ml-1 is used as a control with no mechanical agitation and 10 J ml-1 represents a gentle stimulation, which is suggested not to disaggregate soil structure (Kaiser and Berhe, 2014). Macroaggregates are prone to 50 J ml-1 and 100 to 500 J ml-1 mark the range of microaggregate disaggregation, as many studies stated full disaggregation of soils after application of ∼500 J ml-1 (Kaiser and Berhe, 2014). Larger values were ruled out, although some studies applied energy levels above 500 J ml-1, like Pronk et al. (2011) who could show that silt-sized microaggregates were not dispersed at energy levels ≤800 J ml-1. However, small microaggregates often contain little or no POM (Tisdall, 1996), and energies >710 J ml-1 cause physical damage on mineral particles (Kaiser and Berhe, 2014). Therefore we focus on the range of 0 to 500 J ml-1 as a safe space for the extraction of POM with no other known artifacts."

Line126: why 1% and 0.5%, please justify these amounts [14] 1% is a low but common concentration of POM in soils as well as an amount of MP found in highly contaminated soils (Fuller and Gautam, 2016). We chose these amounts to use the POM economically on one hand and to use on the other hand enough material to find even

small differences of the recovery rate. The use of only 0.5%, alas, is caused by an accident when the measurement had to be applied immediately. However, from our point of view, such slight differences in concentration would not affect the transmission of sound to the POM particles within the slurry. To account net weight differences, our data are in %.

Line 127: If you want to simulate the soil matrix, why did you used only fine sand and not a more heterogeneous mixture? [15] We propose to add the following sentence into Line 127: "We chose a matrix that can be easily suspended by ultrasonication (coarse sand cannot), has a low tendency to coagulate (like clay does), coating (e.g. coating of POM by clay particles, which increases the mean density of the aggregate) and shows a fast sedimentation when the sample is centrifuged. Fine sand, moreover, represents soils that originated from Weichselian sanders or aeolian sand deposition. In this methodical paper, our aim, however, was not to simulate a set of soil textures, but to have a proof of concept to find out if natural or artificial POM is damaged by ultrasonication." and to Line 249: "An exact quantification of the degree of comminution goes beyond the scope of this, because it most probably depends not only on the texture, but also the degree of aggregation and the properties of occluded POM (as differences between forest and farm oLF showed. This will be part of a study in advance to this."

Results

Line 171: two times 100 J/ml [16] The two "100 J/ml" refer to forest oPOM and pyrochar, respectively. We rearranged the sentence to make this more clear: "In sharp contrast, all other samples were decreasingly recovered along with increasing energy levels. Farmland POM, forest POM and pyrochar showed significant differences to the 0 J ml-1 treatment at $\geq$10 J ml-1, $\geq$100 J ml-1 and $\geq$ 100 J ml-1, respectively."

Figure 1 and Table 1 present the same data, so they are redundant, please remove Figure 1 [17] Removed.

For Table 1 and 2, from my prospective, a paired t-test requires independent samples but your samples are not independent (POM forest is from one soil, LD-PE from one plastic film, for example) based on that you can just state an increase or a decreaseor you go for mean values (per energy amount) from farm POM, arable POM, and py-rochar ("natural POM", n = 3) and mean values (per energy amount) from all plasticsamples ("microplastic POM", n = 6, this group could be further subdivided into weath-ered or not weathered), then energy amount or energy amount/ size ditribution can serve as factors in an ANOVA analysis, [18] In this point we disagree with the referee. The 9 materials are independent samples. Both weathered and juvenile PE (e.g.) originated from the same raw material, but were differently treated in advance to the experiment (one was weathered, one not). In consequence, those are different collectives and all variants have 3 replicates and can be compared by use of a paired t-test. The comparison between the energy levels of all variants by an ANOVA is possible but not necessary, as our approach only focus on comparison between one energy level of a certain variant and its 0 J ml-1 control. This is adequately achieved by the t-test.

Captions for Figures 2 a and b: A, B, and C as well as mv need to be explained [19] Done.

Discussion

Lines 181-195: this needs to be discussed in the light of the experimental settings, the artificial soil just contained POM and fine sand, how can these findings be applied to soils with much more clay or iron oxides? [20] We deleted Lines 186-187 ("In consequence, particle size reduction will appear during most ultrasonic treatments aimed to extract oPOMs from soils."). Now the first paragraph is not that bold any more. Further points are mentioned in [15] (texture) and [25] (experimental settings).

Line 197: I do not really see a causal relationship here, please clarify [21] We totally agree that, as we are not yet able to explain the underlying mechanism, causality cannot be stated, but only supposed. We therefore propose to alter Lines 196-198: "The

concurrent decrease of particle size and recovery rate of soil derived POMs and py-rochar and their absence in microplastics might indicate a causal relationship between recovery rate and sensitivity against mechanical stress. The underlying process, how-ever, has not been studied before."

Line 199: this would mean that the fine sand particles form associations with small organic particles but I do not see any evidence for this or a paper cited that describes such phenomena, a reason might be that the density of natural POM is changing be-cause of stronger solubilization processes of smaller particles in density solutions. and Line 203: again, you only have mineral surface of fine sand particles, which are usu-ally less involved in organic matter mineral associations, this needs to be clarified on a mechanistic level using appropriate literature if no own data can be used. and Line 200: needs to be "specific surface area in cm2/g" [22] Thank you very much for this interesting idea. After a new search for literature, we propose to replace the paragraph Line 199-207 with: "We assume a mechanism that prevents POM from detection. This effect appeared in our experiment from energies around 50 J ml-1 with the beginning destruction of oPOM. Sparse data on molecular alteration of organic materials due to ultrasonication showed the transformation of lignin, a major constituent of plant cell walls. One hour of treatment caused the formation of a high molecular weight fraction of about 35% of the lignin content with molecular weights increased by the 450-fold (Wells et al., 2013). This may also increase the density of lignin and ligninoid fractions in soil POM towards the density of the fractionation medium and reduce their recovery rate. In addition, ultrasonication was shown to alter the chemical composition of other dissolved organic matter (DOM, <0.45 $\mu$m) and could lead to the formation of water-soluble molecules and colloids, which become lost during the filtration of the sample (Kaiser and Berhe, 2014)." [23] We also replaced the sentence in Lines 234-237 by: "Their smallest part, fibers and microfragments produced by physical, chemical and biological erosion within the soil, might also be affected by chemical alteration due to both weathering and ultrasonication causing enhanced retention in the HF."

Lines 206-207: why that? please provide more detailed explanations on potential effects on fPOM without any stress from sonication [24] We deleted "and might also occur with small-sized fPOM during density fractionation without application of mechanical stress".

Lines 208-217: again, this is a very general statement but need to be seen in the perspective of your specific experimental settings, and what experiments would be necessary to get more general information [25] We agree with you, that our statements have to be more specific and revised the paragraph in the following way: "As a consequence of the reduction of the recovery rate, farmland, forest and pyrochar POMs remain within a sandy matrix the stronger they are treated by ultrasound. If these findings are applied to ultrasonication/density fractionation of natural soils, not only an increasing number of particle size artifacts can be expected, but also the extraction of occluded POM is increasingly hindered at a certain energy level. After each extraction step, parts of the released oPOM remain within the heavy fraction, a carry-over artifact. This leads to an underestimation of the extracted oPOM fractions and an overestimation of the mineral-associated organic matter fraction (MOM), that natural part of the soil organic matter (SOM), which is adsorbed on mineral surfaces of the heavy fraction and mainly assumed to be molecular. According to our data, a reduction of recovery rates would appear at 10 J ml- 1 in farmland soils and 100 J ml-1 in forest soils as well as at 100 J ml-1 when extracting pyrochar particles. Thus, the artifact would affect the extraction of oPOM from microaggregates of all samples and also the extraction of oPOM from macroaggregates in farmland soils. However, further research has to elucidate, if these results can be applied to natural soil samples."

Line 222: define "phenomenal influence" [26] "phenomenal" deleted.

Lines 218-225: again, any recommendations how such effects could be minimized during fractionation. [27] Unfortunately, we don't have. We propose to add after Line 225: "In respect to coming experiments, comminution and reduced recovery rate of the oPOM can possibly be avoided by not exceeding the energy levels mentioned here – or

by determining a specific energy cut-off for each natural soil in preliminary studies. Regarding the application of higher energy levels, detailed investigation on the underlying mechanism are necessary to give such recommendations."

Line 226: again, very general statement, define "plastic" [28] We added: "Microplastic particles, whether they are weathered following DIN ENISO4892-2/3 or pristine, are ..."

Line 227: I recommend to avoid statements like "no carry-over", for such a bold statement the data are not sufficient [29] We replaced the "no" by "significantly less".

Conclusion "... fractions only extractable with higher energy levels or were bound to ..." (Line 246) and "... at the mineral phase..." (Lines 250-251) deleted.

Best regards,

Dr. Frederick Büks, M.Sc. Gilles Kayser, M.Sc. Antonia Zieger, Prof. Dr. Friederike Lang, Prof. Dr. Martin Kaupenjohann

Additional references

Andrady, A.L.: The plastic in microplastics: a review, Mar. Pollut. Bull., 119, 12–22, https://doi.org/10.1016/j.marpolbul.2017.01.082, 2017

Bläsing, M. and Amelung, W.: Plastics in soil: Analytical methods and possible sources, Science of The Total Environment, 612, 422–435, https://doi.org/10.1016/j.scitotenv.2017.08.086, 2018.

Braunack, M., Hewitt, J. and Dexter, A.: Brittle fracture of soil aggregates and the compaction of aggregate beds, Journal of Soil Science, 30, 653–667, https://doi.org/10.1111/j.1365-2389.1979.tb01015.x, 1979.

Breitbach, M., Bathen, D., Schmidt-Traub, H. and Ebener, H.: Stability of adsorber resins under mechanical compression and ultrasonication, Polymers for Advanced Technologies, 13(5), 391-400, https://doi.org/10.1002/pat.203, 2002.

Büks, F., Loes van Schaik, N., and Kaupenjohann, M.: What do we know about how the terrestrial multicellular soil fauna reacts to microplastic?, SOIL, 6, 245–267, https://doi.org/10.5194/soil-6-245-2020, 2020.

de Souza Machado, A. A., Lau, C. W., Till, J., Kloas, W., Lehmann, A., Becker, R. and Rillig, M. C.: Impacts of Microplastics on the Soil Biophysical Environment, Environmental Science & Technology, 52, 17, 9656–9665, https://doi.org/10.1021/acs.est.8b02212, 2018.

Fuller, S. and Gautam, A.: A procedure for measuring microplastics using pressurized fluid extraction, Environ. Sci. Technol., 50, 5774–5780, https://doi.org/10.1021/acs.est.6b00816, 2016.

Kale, S. K., Deshmukh, A. G., Dudhare, M. S., and Patil, V. B.: Microbial degradation of plastic: a review, J. Biochem. Technol., 6, 952–961, 2015.

Pronk, G. J., Heister, K., and Kögel-Knabner, I.: Iron Oxides as Major Available Interface Component in Loamy Arable Topsoils, Soil Science Society of America Journal, 75(6), 2158, https://doi.org/10.2136/sssaj2010.0455, 2011.

Rezaei, M., Riksen, M. J., Sirjani, E., Sameni, A., and Geissen, V.: Wind erosion as a driver for transport of light density microplastics, Sci. Total Environ., 669, 273–281, https://doi.org/10.1016/j.scitotenv.2019.02.382, 2019.

Tisdall, J.: Formation of soil aggregates and accumulation of soil organic matter, Structure and organic matter storage in agricultural soils, 57-96, 1996.

Wells, T., Kosa, M., & Ragauskas, A. J.: Polymerization of Kraft lignin via ultrasonication for high-molecular-weight applications, Ultrasonics Sonochemistry, 20(6), 1463–1469, https://doi.org/10.1016/j.ultsonch.2013.05.001, 2013.

Zubris, K. A. V., and Richards, B. K.: Synthetic fibers as an indicator of land application of sludge, Environmental Pollution, 138(2), 201–211, https://doi.org/10.1016/j.envpol.2005.04.013, 2005.

---

## Referee Comment (RC2) · Anonymous Referee #2 · 14 Sep 2020

General comments: This paper reports a study on recovery rate and change in size distribution of soil oPOM, pyrochar, and microplastics under a series of ultrasonication treatments. It is shown that while ultrasonication is widely used in soil particle fractionation, it may cause artifact and false estimation of POM stability even with low power. For microplastics, both recovery rate and size distribution are not significantly changed by ultrasonication treatment. The results are valuable for both soil aggregate-related researches and microplastics studies. However, the study needs some essential data to support its conclusion and explanation of the results.

Specific comments: 1. For oPOM and pyrochar, the recovery rate decreased with the

increment of ultrasonication power. The cause was supposed to be an increase of new active surface to absorb the comminuted oPOM after disintegration of soil aggregates. However, this explanation is unlikely applicable to pyrochar. 2. As mentioned in Ince (2001) and confirmed in Kaiser & Berhe (2014), ultrasonication induced high temperature may reduce total C content due to oxidative reactions. If this happens, the conclusion of "counting up to around 36.2 to 64.2 % of POM to the MOM" is really overestimated. I would like to know how much oPOM was lost and how much was transferred to MOM in this study. 3. In line 149, "About 100 mg POM were suspended" for particle size analysis. However, the initial quantity of oPOM in each vessel is 20 g * 0.5% = 100 mg. Therefore, with a recovery rate may be as low as 50%, it is unlikely to get 100 mg of oPOM for particle size analysis. 4. The farm and forest soils used for this experiment were from an organic horticulture and a spruce/beech mixed forest. However, soil organic C content was only 4.9 and 7.3 g kg-1. Please check these data. 5. Is the weight of POM measured or the C content measured? 6. There are some grammar errors, including explanation of the calculation of CF.

---

## Author Comment (AC2) · 23 Oct 2020

Dear referee #2.

Many thanks for your proofreading and your helpful comments. We added the requested data to our manuscript and supplements. In the following, we want to explain how we propose to improve our explanation as in your favor.

Comment 1: For oPOM and pyrochar, the recovery rate decreased with the increment of ultrasonication power. The cause was supposed to be an increase of new active surface to absorb the comminuted oPOM after disintegration of soil aggregates. However,

this explanation is unlikely applicable to pyrochar.

[1] We hope that we understood your comment correctly. Pyrochar has an enormous internal surface, but we assume that also pyrochar receive a larger outer surface if particles are comminuted. However, based on the comment of referee #1 on the lack of evidence in literature regarding the association of small organic particles with sand grains we refraine from this explanation and instead added reply [22] to referee #1.

Comment 2: As mentioned in Ince (2001) and confirmed in Kaiser & Berhe (2014), ultrasonication induced high temperature may reduce total C content due to oxidative reactions. If this happens,the conclusion of "counting up to around 36.2 to 64.2 % of POM to the MOM" is really overestimated. I would like to know how much oPOM was lost and how much was transferred to MOM in this study.

That is a very interesting question, which is really improving our work. We did measurments in this regard and added after line 164 into the material & methods section: "2.5 organic matter balance: A second set of triplicates of pyrochar and farm soil oPOM were treated similarly at 0 and 500 J/ml to balance the complement of the recovered POM. For this purpose, the C concentration within the lyophilized sediment was measured by use of a CNS analyzer and converted to POM mass by use of the C content (%) of the respective organic matter. In addition, the mass gain of the cellulose acetate filters was measured after rinsing the sample and drying the filter at 70°C for 24 hours. The DOC concentration of the filtrate was measured and converted to DOM by use of an assumed 50% C content. The difference of these and the recovered fractions compared to the initial weight of organic particles is termed the balance loss during the extraction procedure."

Corresponding to that, we added the following to the resuls section after Line 179: "3.3 Mass loss: The treatment of pyrochar triplicates with 500 J/ml resulted in a recovery rate of $54.3\pm5.2$ % after density fractionation. In turn, $34.9\pm3.7$ % of the POM remained in the sediment, $0.6\pm0.1$ % into the DOM fraction and <0.5 % onto the filter,

leading to a balance loss of 10.2±2.1 % (Fig. 2). The respective data of farm oPOM are 54.6±1.9 %, 20.3±3.1 %, 5.1±0.2 %, <0.5 % and 20.0±1.5 %. Samples treated with 0 J/ml instead showed a significantly higher recovery rate and lower retention compared to the 500 J/ml samples. In contrast, the balance loss remained constant between 0 and 500 J/ml." The data are shown in an additional figure and added to the supplements.

We furthermore supplemented our comment [22] to referee #1 in the discussion section as follows: "We assume a mechanism that prevents POM from detection. This effect appeared in our experiment from energies around 50 J ml-1 with the beginning destruction of oPOM. As mentioned in Ince et al. (2001) and confirmed in Kaiser and Berhe (2014), ultrasonication induced high temperature may reduce total C content due to oxidative reactions, but the balance loss, constant between 0 and 500 J/ml in both pyrochar and farm oPOM, implies that there is no burning of organic matter due to ultrasound treatment. The recovery rate decreases in the same degree as the retention in the sediment increases when ultrasound is applied, while relics on the filter and lost DOM, which doubled on a low level, play a minor role. Extreme thermal conditions occuring during ultrasoincation, however, may explain the retention of POM within the sediment increasing with higher energy levels. Sparse data on molecular alteration ..."

Comment 3: "About 100 mg POM were suspended" for particle size analysis. However, the initial quantity of oPOM in each vessel is 20 g* 0.5% = 100 mg. Therefore, with a recovery rate may be as low as 50%, it is unlikelyto get 100 mg of oPOM for particle size analysis.

[5] We are sorry for this phrase has escaped our notice. It actually means "up to 100 mg" and refers to the plastic samples, which had an initial weight of 0.2 g and were recovered by nearly 100%. For pyrochar and the oPOMs the QicPic used a smaller amount according to the extracted matter. The actual range of sample weight is therefore "30 to 100 mg", which is to correct in line 149.

Comment 4: The farm and forest soils used for this experiment were from an organic horticulture and a spruce/beech mixed forest. However, soil organic C content was only 4.9 and 7.3 g kg-1. Please check these data.

[6] Thank you for your mindful reading. It is indeed 4.93% and 7.32% (or 49.3 mg/kg and 73.2 mg/kg) and will be corrected in Lines 100 and 102.

Comment 5: Is the weight of POM measured or the C content measured?

The recovery rates base on POM weights. That is because (1) this work focus on mass losses and (2) C analytic is destructive and would have doubled the operational effort with respect to the following particle sizing.

Comment 6: There are some grammar errors, including explanation of the calculation of CF.

We thoroughly reread our manuscript and corrected some grammatical errors that had escaped our notice.

Best regards,

Dr. Frederick Büks, M.Sc. Gilles Kayser, M.Sc. Antonia Zieger, Prof. Dr. Friederike Lang and Prof. Dr. Martin Kaupenjohann